# Impacts of Topology and Bandwidth on Distributed Shared Memory Systems

Jonathan Milton  and Payman Zarkesh-Ha *

Department of Electrical and Computer Engineering (ECE), University of New Mexico, Albuquerque, NM 87131-1070, USA; jmilton@unm.edu
* Correspondence: pzarkesh@unm.edu

**Abstract:** As high-performance computing designs become increasingly complex, the importance of evaluating with simulation also grows. One of the most critical aspects of distributed computing design is the network architecture; different topologies and bandwidths have dramatic impacts on the overall performance of the system and should be explored to find the optimal design point. This work uses simulations developed to run in the existing Structural Simulation Toolkit v12.1.0 software framework to show that for a hypothetical test case, more complicated network topologies have better overall performance and performance improves with increased bandwidth, making them worth the additional design effort and expense. Specifically, the test case HyperX topology is shown to outperform the next best evaluated topology by thirty percent and is the only topology that did not experience diminishing performance gains with increased bandwidth.

**Keywords:** high-performance computing; distributed shared memory; topology; structural simulation toolkit; optimization

## 1. Introduction

The need to develop and operate high-capability supercomputers is recognized in the High-Performance Computing (HPC) community as a prerequisite for the advancement of a number of scientific and technical challenges, ranging from basic science to "big data" operations [1]. As the state-of-the-art transitions into exascale, designing and building supercomputers becomes increasingly difficult. On top of the challenges associated with scalability and raw performance, cost, power consumption [2], cooling, reliability [3], and programmability [4] all need consideration. Overcoming design challenges requires an integrated approach to how systems are architected. Changes to processors, memory, and networks need to be assessed concurrently to properly capture the complex interactions between them.

The design of the memory architecture plays a critical part in enhancing the capabilities of local nodes in an HPC system. Integrating processing cores with memory provides significant node performance advantages [5]. Furthermore, related works have found the network design has a major role in overall performance with many high bandwidth architectures available to choose from [6]. As HPC problems try to optimize localization, even in real time [7], the endpoint network capabilities must be able to handle the heavy workload without becoming a bottleneck. As cores scale into the tens of thousands [8], the network must improve or become the limiting factor for performance [9]; further defining this limit point is an objective of this paper.

While there are many factors to consider, memory architecture is one that greatly influences many other HPC design decisions. Many areas of shared memory architecture have been thoroughly researched in an attempt to determine the optimal configuration, but there is a gap in the research dealing with the trade-offs between bandwidth and topology for distributed shared memory architectures and their impact on performance for different

workloads. This work will address that gap by outlining a process to determine the balance of design parameters that offers the best average performance across the expected workload for an HPC system—more specifically, identification of the optimal network configuration for a specific software type to minimize the overall execution time.

As it is impractical to build prototypes to fully explore the design space of an HPC system, simulation must be used to evaluate the available options. Historically, there has been no single point solution for performing these evaluations. While simulators exist for individual components, there have not been good options for uniting them together to evaluate the system as a whole until the development of the Structural Simulation Toolkit [10].

Simulation will be used in this work to determine quantitatively the ideal network topology and network bandwidth for small 16-node and 256-node architectures, the results of which will be extended to a larger number of nodes. To limit the scope of the problem, only performance will be evaluated.

### 1.1. Description of the Problem

For every HPC system using distributed shared memory, there is an eventual decline in return on investment for increasing the network bandwidth beyond a certain capacity. At some point there will be sufficient bandwidth and the limiting factor for data transfer will transition to another aspect of the design—with a hard limit driven by the CPU rate of request issuances. This paper seeks to identify that point using simulation for a variety of network topologies, verifying the concept and listing what correlations, if any, are found.

The concept of variable bandwidth has been explored before in the context of a single topology to look at the potential for cost savings in hardware and energy for reducing the number of links between routers [11]. A very detailed analysis exploring the impacts of multiple variables on performance, including the impact of reducing bandwidth to the peak performance of several topologies, also exists, but does not address the question of optimization or look at the potential for performance gains [12].

This work differs in that the multivariable analysis is conducted across topologies and bandwidth together, with the objective being identification of performance gains rather than negative effects. It also explores the rate of change to performance improvement in a way not addressed by other studies. This question has likely been left unaddressed as commercial hardware has a specified bandwidth and the issue of reducing operating costs due to power consumption has only more recently become critical [2]. As technology advances to integrate more pieces of the architecture into a single piece of hardware, the question of when it stops making sense to increase performance on one particular aspect becomes more critical as it will be taking resources away from other aspects of the design. Being specific to a distributed shared memory space with direct access to the memory from the network is another aspect differentiating this from related works.

Several topologies have been selected to cover a range of design complexities. Mesh and Torus are good representations of simple and physically scalable HPC network solutions [13]. Dragonfly and HyperX are more complex options selected for being implementable with uniform link bandwidths but having similarities to the common Fat-Tree architecture, including the ability to implement Fat Tree networks based on the configuration parameters [14,15]. Standard Fat-Tree architecture has been excluded from comparison as the implementation requires increasingly higher network performance at the top with scaling and would not be a good direct comparison with the other architectures [16,17]. For the uniform distribution of traffic between nodes in this simulation, there are no advantages of using that topology [16]. The Dragonfly and HyperX topologies evaluated here are in many ways similar to a Fat Tree, but have the benefit of eliminating some latency points through bypassing higher levels of the tree while maintaining a fairly consistent distribution of bandwidth between nodes [15]. An infinite radix Single Router (Crossbar or Star topology equivalent) bounds performance on the high end, and a Ring—as an amusing

representation of probably the worst possible architecture for HPC—serves as the lower performance bound. For each topology, several different memory bandwidths are assessed.

### 1.2. Simulation Configuration

The simulations developed for this evaluation consist of nodes with 4 processing cores each; the cores have a 256 B private cache, attached to a 512 B L2 cache which is shared across the 4 cores (the caches are intentionally made very small to force turnover during simulation runs). The small node counts are driven by the need to complete the simulations in a timely manner without requiring parallel computing resources. Each node has 1 GB of network attached memory for which the bandwidth (both local and to the network) starts at 1 GB/s and is doubled in incremental steps up to 8 GB/s. In general, each endpoint node has a dedicated router; the exceptions being the Single Router topology out of necessity, and the HyperX topology to keep the group size the same as Dragonfly. The execution model for each core is simply a stream of random read operations to the memory space. The random memory requests drive approximately uniform traffic levels between nodes and across links, which is one of the typical traffic patterns used for topology simulations [11,18] and is similar to many applications of a properly optimized and balanced parallel HPC workload [19]. While this configuration is not truly representative of an HPC architecture, it will allow simulations to be completed in a reasonable amount of time, demonstrate the feasibility of the analysis, and give insight into the performance of other systems in a more practical HPC platform.

### 1.3. The Structural Simulation Toolkit

The Structural Simulation Toolkit (SST) [10] software was selected to implement the simulations of this work as it is a discrete event simulator developed jointly by a number of institutions to address this gap and provide a simulation framework covering everything from System on a Chip to large-scale HPC systems, even allowing for modeling of impacts down to the transistor level [20]. The core framework provides timekeeping, event exchange control, and statistics while individual element libraries are used to implement system components [10]. SST allows for parallel simulation of hundreds of thousands (or more) nodes in sufficient detail to accurately assess the design space of an HPC system [10]. SST joins models for processors, memory, networks, and more into a single simulation [10].

The simulation framework of SST [10] builds on a long history of architectural and network simulators such as NS-3 [21], M5 [22], and A-SIM [23]. Additionally, it builds on power dissipation modelling [24,25] and integrates many existing simulators as plug-in modules without requiring additional code revisions. SST works by including individual component models in a scalable, parallel, and open-source framework [10].

## 2. Materials and Methods

The source code and data presented in this study are available on request from the first author. SST source code is available from https://sst-simulator.org/ (accessed on 13 April 2023). Example source code from the SST tutorials illustrating how to develop simulations can be obtained at https://github.com/sstsimulator/sst-tutorials (accessed on 13 April 2023). The individual topology simulation source code created for this work is not publicly available at the time of publication due to forming part of a larger body of work to be made publicly available at a later date via https://digitalrepository.unm.edu/ (accessed on 13 April 2023).

### 2.1. The Structural Simulation Toolkit

For this project, exploration of computer system design using modeling of the complex interactions between processor, memory, and network was performed using SST, an open, modular, parallel, multi-criteria, multi-scale simulation framework [10]. SST was developed to explore innovations in both programming models and hardware implementation of highly concurrent systems and utilizes a modular design to allow extensive exploration of

system parameters while maximizing code reuse, and provides an explicit separation of instruction interpretation from microarchitectural timing [10]. This is built upon a high-performance hybrid discrete event framework. SST can handle highly concurrent designs where the instruction set architecture, microarchitecture, and memory interact with the programming model and communications system [10]. The package provides two novel capabilities: The first is a fully modular design that enables extensive exploration of an individual system parameter without the need for intrusive changes to the simulator [10]; the second is a parallel simulation environment based on Message Passing Interface [10]. This provides a high level of simulation performance and the ability to evaluate large systems using parallel processing techniques.

*2.2. Topology Simulations*

Using a simple model, this project explored, demonstrated, and evaluated the ability of SST [10] to optimize a design, by analyzing the tradeoff in resource allocation between bandwidth and topology for multicore shared memory systems. The specifics of the system were determined by first evaluating a four-processor node of quad-core processors. Each core contains a private L1 cache, and the L2 cache is shared across all four cores. In total, 1 GB of memory is attached to each processor; the address space is shared and spanned, but not interleaved. The model accounted for an onboard network between the processors with direct memory access into the shared spaces, while a directory controller for each space maintained the coherency. From here, cache size, on-chip network rates, and other parameters were optimized prior to exploring the impacts of bandwidth and topology.

At this point, a model for each topology was constructed using python scripts to feed into SST [10]. The models consist of three overall sections: The first section consists of approximately 300 configuration variables which control all aspects of the design. These were implemented this way to enable future flexibility and automation as well as to ensure parameter consistency throughout the design. The second section uses recursive scripting to instantiate an arbitrary number of nodes, containing and arbitrary number of processors, containing an arbitrary number of cores. This section also creates all the other components present on the node or in the processor and links them together. Updating the node, processor, or core specifics not controlled by variables only needs to occur in one place. These first two sections are common across all topology models. The final section creates the network topology, instantiating routers and linking everything together. This was also written with recursive scripting to allow maximum flexibility in reconfiguration. For example: the Torus simulation defines the dimensions, shape, and links in each dimension using variables so the 4D $2 \times 2 \times 2 \times 2$ torus previously evaluated could quickly be reconfigured to 3D $2 \times 2 \times 4$ for comparative analysis.

In executing the simulations, the bandwidth was varied for each trial while using a constant random pattern seed to ensure results between trials were truly comparable. Initial results are presented here while ongoing simulation trials explore additional memory access patterns and bandwidth rates.

Each topology evaluated includes a network map for the 16-node configuration using images of a rack mount server to represent each node (group of four processing cores with an attached memory), along with routers and lines for the interconnecting links. The performance improvement as bandwidth is increased is plotted as speedup relative to initial performance at 1 GB/s.

2.2.1. Single Router (Crossbar) Topology

The Single Router configuration (Figure 1) has all the nodes tied to the same router and represents the near theoretical upper limit in performance for the analysis as each node has a direct link to every other node's memory through a fully switched network crossbar. The primary drawback to this configuration is that it is not physically scalable to large numbers of nodes. Physical design constraints of a 1U rack space limit readily available hardware to around 48 ports. Growing beyond this would require additional high bandwidth links

between multiple switches, at which point the design has transitioned to another topology. Every connection in this case is only two hops away. Performance improvement for this configuration is shown in Figure 2.

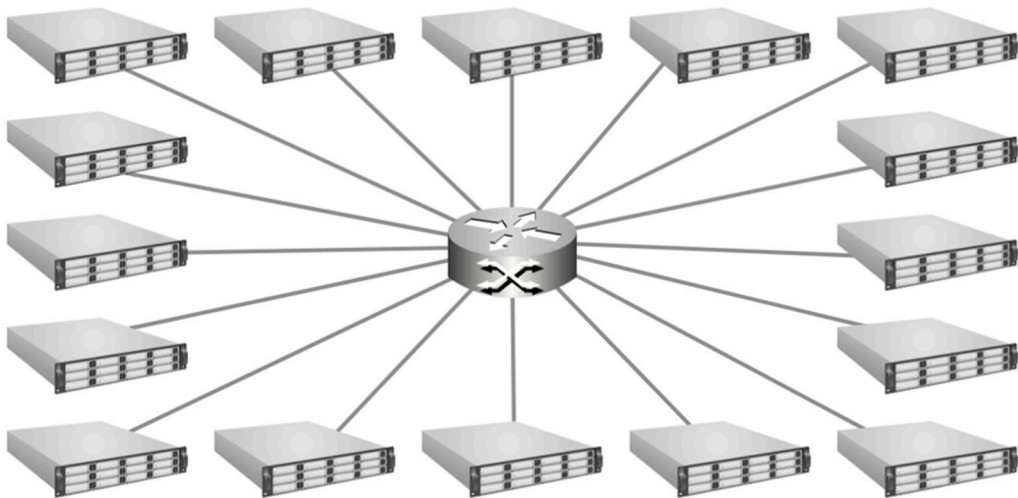

**Figure 1.** Single Router Topology.

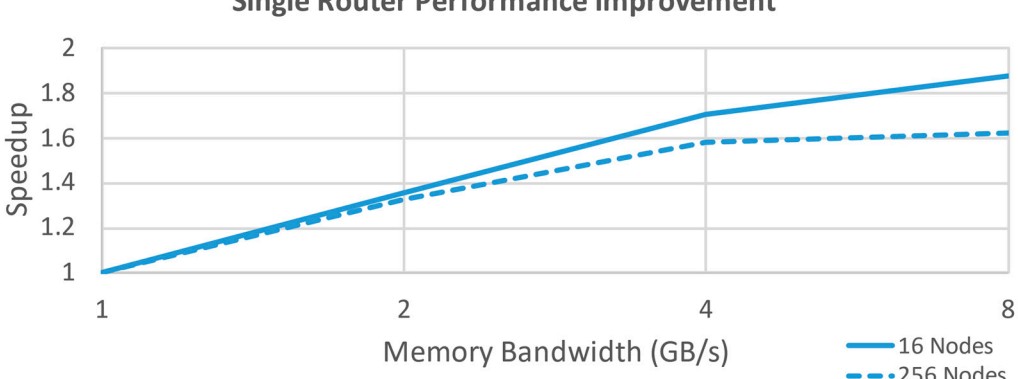

**Figure 2.** Single Router Performance.

### 2.2.2. Ring Topology

The Ring configuration (Figure 3) links each node's router to only two neighbors in a large loop, such that there are likely multiple hops to reach the desired memory address. While easily scalable as far as physical construction goes, it does not scale well for performance considerations with a large number of nodes. Network performance quickly becomes a bottleneck as each link needs to pass a large amount of traffic to nodes farther along the ring. For 16 nodes, the maximum one-way travel distance for information is eight hops. Performance improvement for this configuration is shown in Figure 4.

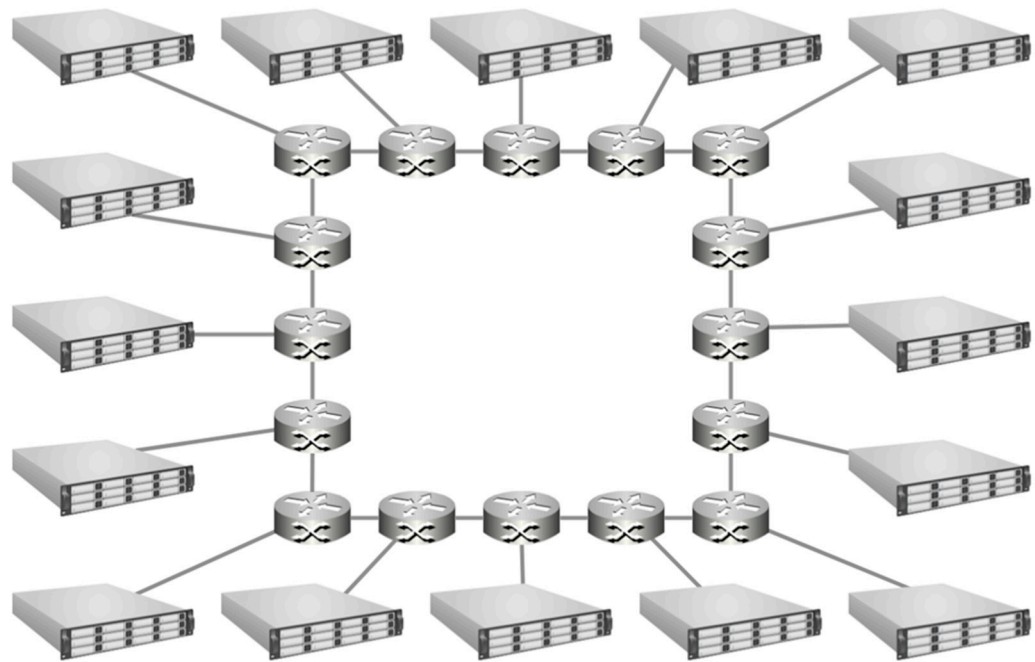

**Figure 3.** Ring Topology.

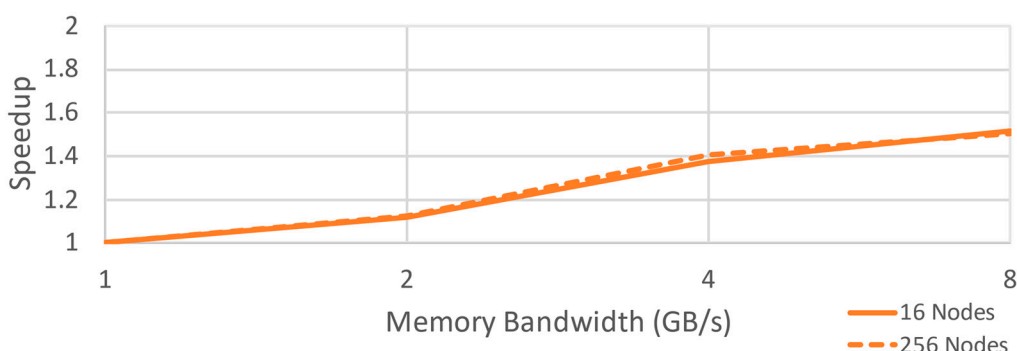

**Figure 4.** Ring Performance.

2.2.3. Mesh Topology

The Mesh configuration (Figure 5) links each node's router to two or four neighbors in a two-dimensional grid pattern such that there are fewer hops to reach the desired memory address. Again, this is easily scalable as far as physical construction goes, but it does not scale well for performance considerations with a large number of nodes. The maximum number of hops for 16 nodes has been reduced to 6 by changing to this topology. Performance improvement for this configuration is shown in Figure 6.

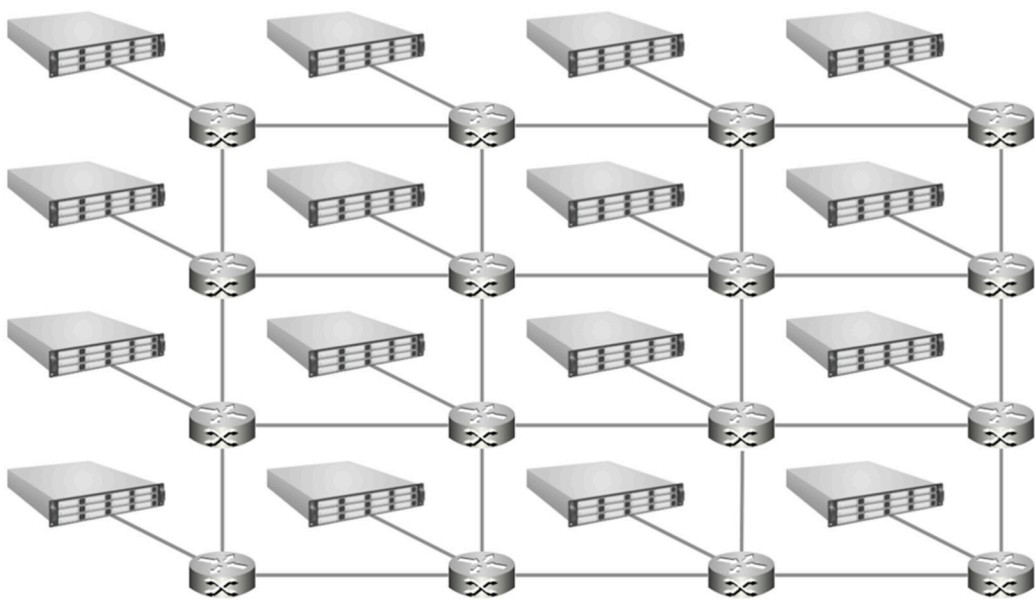

**Figure 5.** Mesh Topology.

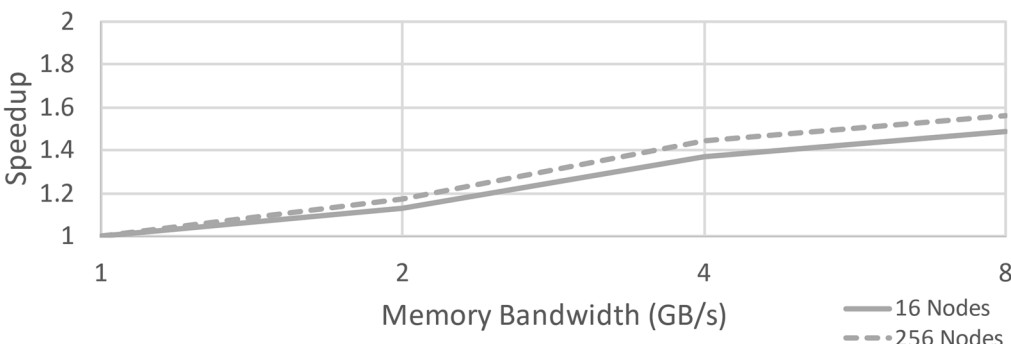

**Figure 6.** Mesh Performance.

2.2.4. Torus Topology

The Torus configuration (Figure 7) used is 4D ($2 \times 2 \times 2 \times 2$ for 16 nodes, $4 \times 4 \times 4 \times 4$ for 256 nodes), which links each node's router to eight neighbors. This substantially reduces the number of hops required to reach the desired memory address (note that for the 16-node case each neighbor is counted twice, effectively doubling the bandwidth; the loops around links which would be present in a 4D Torus with more nodes are not illustrated here). Figure 7 uses a unique color for each dimension to highlight the individual links, yellow for X, green for Y, orange for Z, and blue for W (the compute nodes have been omitted to improve readability; there is one node attached to each router). This topology is straightforward to improve performance through increasing the number of dimensions, and flexible to accommodate different physical configurations. This is not as easily scalable as far as physical construction goes, but it does scale fairly well for performance considerations to a large number of nodes. The maximum number of hops for a 16-node design has dropped to four and the effective bandwidth along those hops is doubled for this specific example. Performance improvement for this configuration is shown in Figure 8.

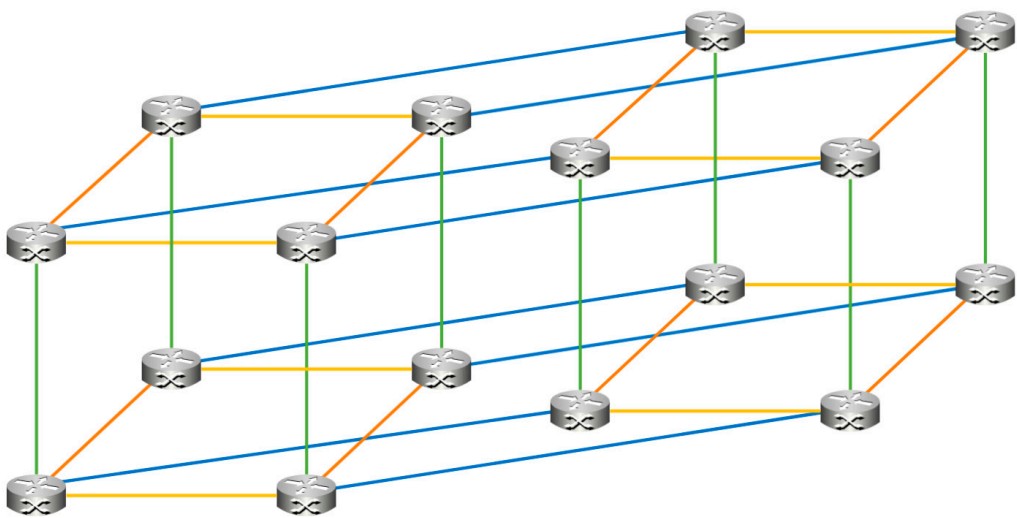

**Figure 7.** Torus Topology.

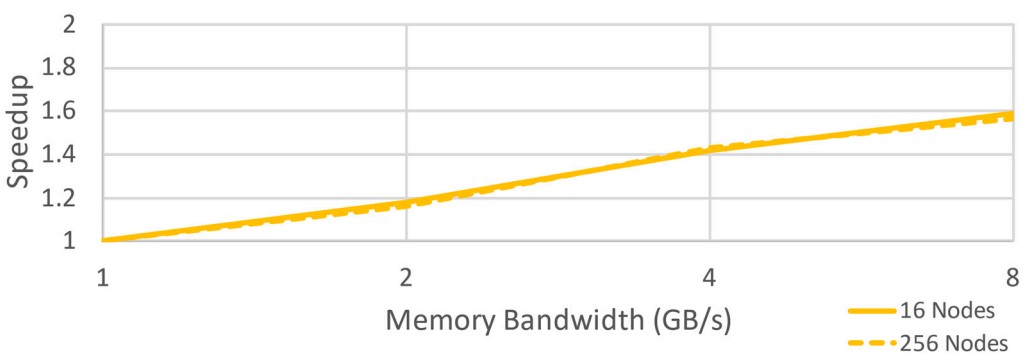

**Figure 8.** Torus Performance.

### 2.2.5. Dragonfly Topology

The Dragonfly configuration (Figure 9) used for 16 nodes is a slightly modified 4, 1, 1, which places each node's router into one of 4 groups of 4 fully meshed neighbors per group. For 256 nodes, there are 16 groups of 16. Each router in the group connects to one other group (excepting one router, whose connection would be a parallel link), the result being a fully connected network with at most three hops (at any scale) to the desired memory address and typically a choice of more than one path if there is network congestion. Figure 9 highlights the groupings and shows the intergroup links for the topology (the compute nodes have been omitted to improve readability, there is one node attached to each router). This specific design effectively implements a three-level Fat Tree, using only two levels with uniform hardware requirements. Again, it is not as easily scalable as far as physical construction goes, but it does scale nicely for performance considerations to a large number of nodes. Performance improvement for this configuration is shown in Figure 10.

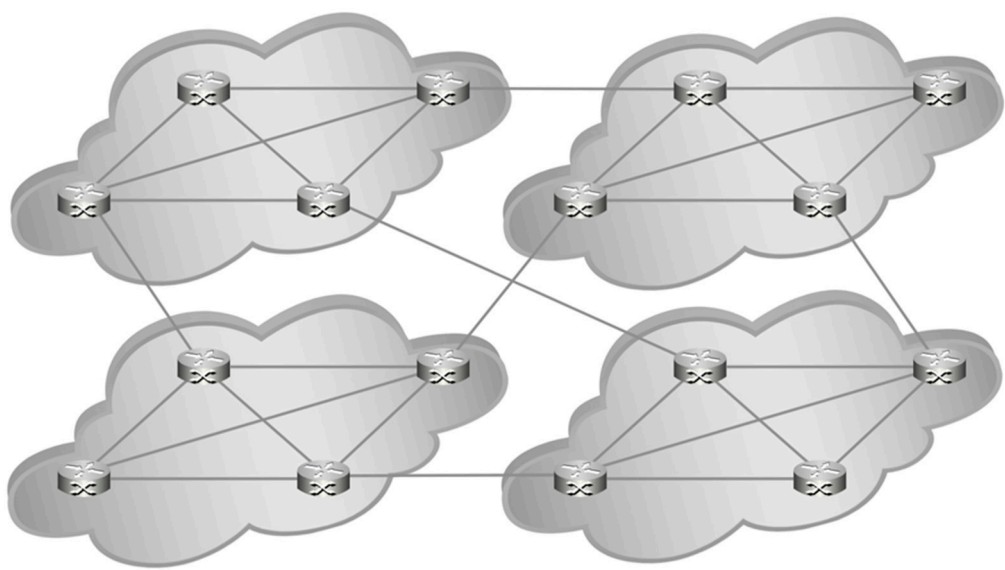

**Figure 9.** Dragonfly Topology.

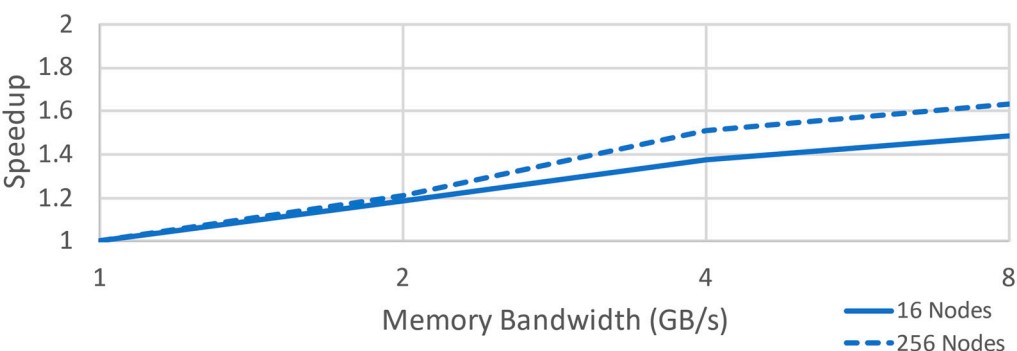

**Figure 10.** Dragonfly Performance.

### 2.2.6. HyperX Topology

The HyperX configuration (Figure 11) used for 16 nodes is a 4 × 2, which places each endpoint node into groups of two that are connected together on a single router. That router connects to every other router in its dimension in a full mesh, in this case three links for the first dimension and one link for the second. For 256 nodes, 4 × 32 was used. This results in a fully connected network with at most four hops to the desired memory address and several choices for path if there is network congestion. Figure 11 highlights the grouping and shows the intergroup links for the topology; note that HyperX is the only topology other than single router with two nodes per router. This is another implementation of a three-level Fat Tree using only two levels and uniform hardware requirements. Once more, it is not as easily scalable as far as physical construction goes, but it does scale nicely for performance considerations to a large number of nodes and can be modified for simpler physical construction by not fully meshing at a penalty of increased hops. Performance improvement for this configuration is shown in Figure 12.

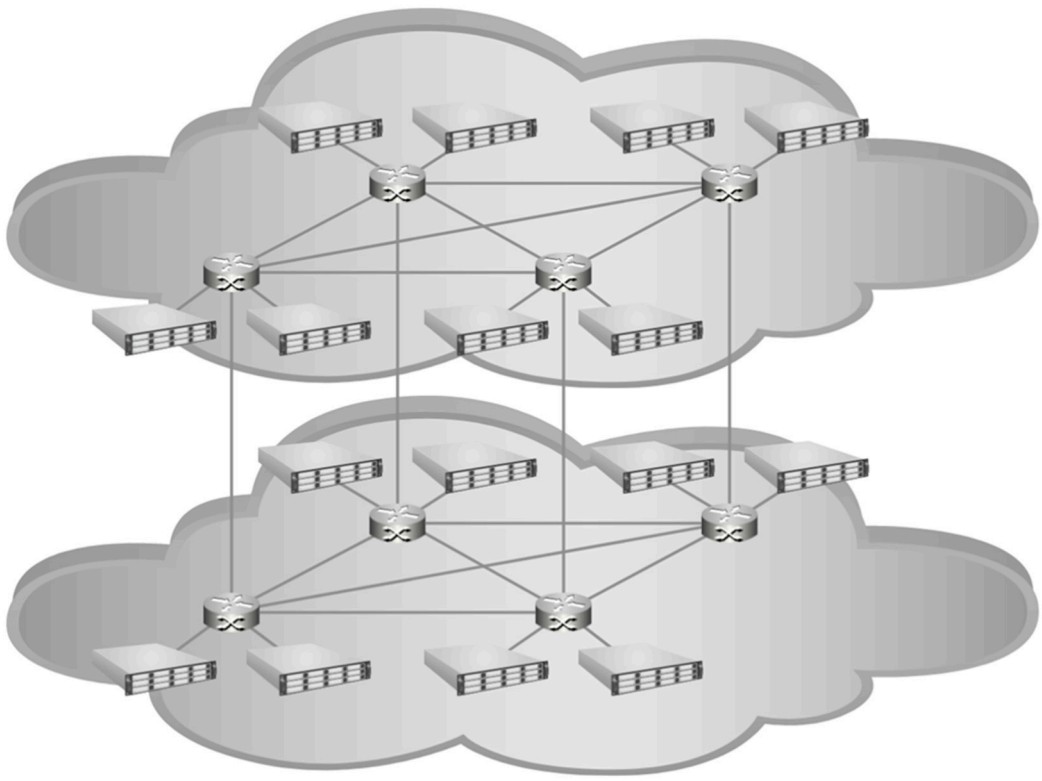

**Figure 11.** HyperX Topology.

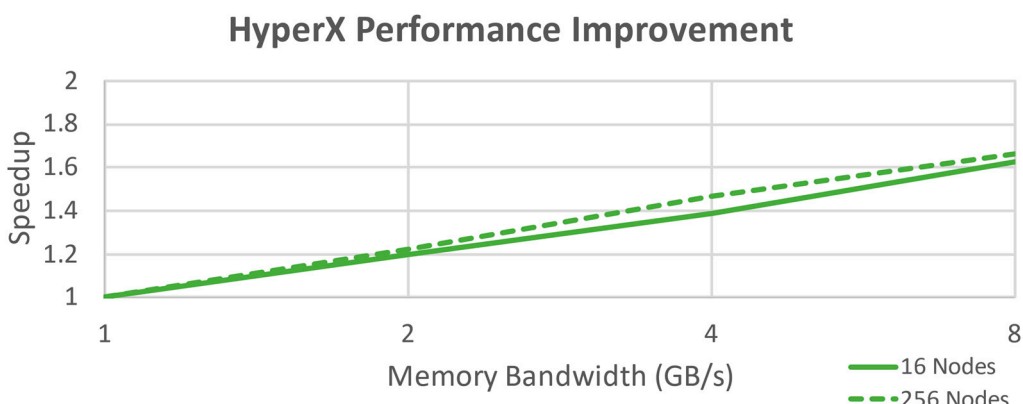

**Figure 12.** HyperX Performance.

### 3. Results

Combined results for the simulations on 16 nodes (Figure 13) indicate that performance gains begin to taper off for most cases in the bandwidth range from 4 GB/s to 8 GB/s and that this area likely provides the best performance return on investment for this specific hypothetical hardware definition before the speedup starts to flatten out; this becomes even more evident when the speedup plot is taken out of log scale. For 256 nodes (Figure 14), there is a more dramatic decline in returns after 4 GB/s for all the topologies except HyperX, which is still increasing at a reasonable rate. As the possible bandwidth use driven by the maximum node memory request rate is still well above this level, another aspect of the design is now the performance bottleneck. Detailed analysis of the performance statistics generated by SST [10] indicate the cause is a combination of latency and the limit on allowed outstanding requests. Identification of a good design starting point for the network bandwidth has been achieved, and the focus in this case should shift towards improving

the latency and evaluating the impact of increasing the outstanding request limit before further increasing the bandwidth.

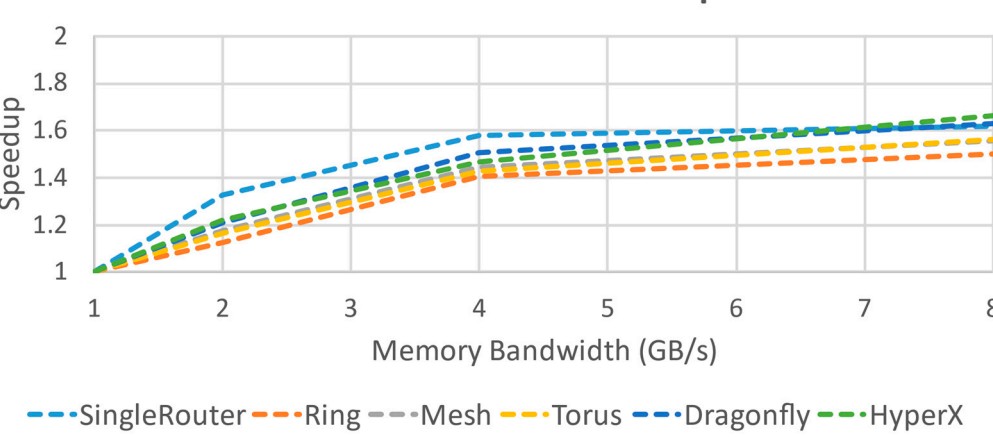

**Figure 13.** Combined Performance Improvement for 16 Nodes.

**Figure 14.** Combined Performance Improvement for 256 Nodes.

Plotting the performance of all 16 node topologies against the theoretical limit (Figure 15) shows that in this case the HyperX topology not only benefits the most from increased bandwidth but is also the top performer in terms of compute time; followed by Torus and Dragonfly, with almost identical performance. HyperX does not have an advantage in terms of maximum hop count against Torus or Dragonfly topologies in this test case, but the high number of paths creates an increased effective bandwidth that gives it a performance boost here. For 256 nodes (Figure 16), the topologies fall into the same order, except that Dragonfly is now showing improved performance against Torus. The potential number of hops for Torus grow with the node count so it is not keeping up with the better optimized HyperX and Dragonfly. The performance compared to the hypothetical limit from the Single Router topology is about half of what it was for 16 nodes; using an infinitely scalable unlimited radix router as a benchmark is likely causing this drop. It is not surprising that the ring topology has the weakest performance of all the architectures simulated.

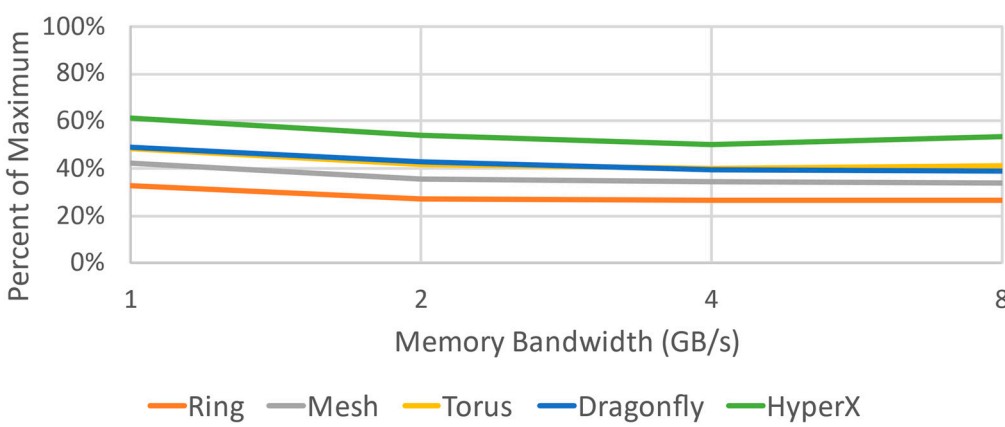

**Figure 15.** Relative Performance of Topologies for 16 Nodes.

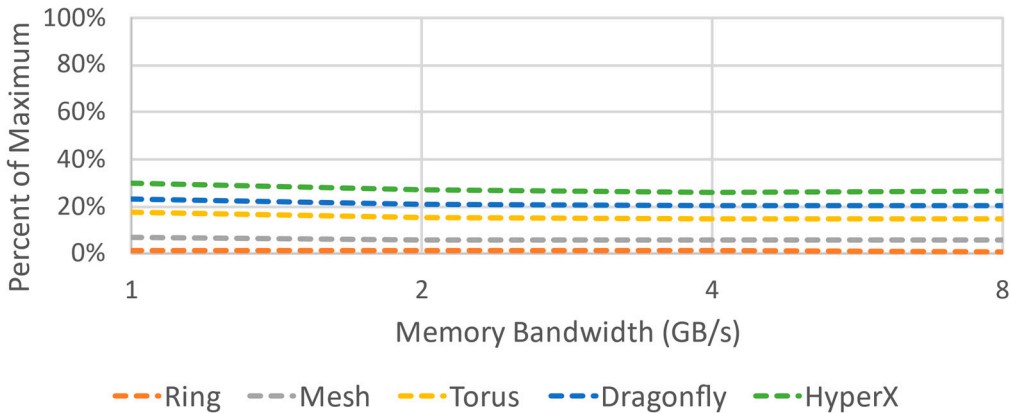

**Figure 16.** Relative Performance of Topologies for 256 Nodes.

### 4. Discussion

The results of this analysis indicate that for the hypothesized fixed design parameters, a HyperX topology with 8 GB/s memory bandwidth provides the best overall performance against other potential topologies (excluding the unscalable Single Router topology). Expected performance gain in this system for using HyperX is a 20–25% increase for both node counts over Dragonfly, the next best-performing topology. Against Torus at 256 nodes, the increase is 40–45%. Performance gains of 19–25% are also achieved each time the bandwidth is doubled for HyperX for either node count, whereas Dragonfly and Torus have a broader and less predictable improvement spread of 10–26%. HyperX was also the only topology evaluated that was not experiencing significant diminishing returns by 8GB/s; even the theoretical limit of the Single Router was dropping off at that point.

To better understand the scalability of the model closer to actual HPC specifications, the HyperX simulation was increased to 2048 nodes and evaluated for a 32, 64, and 128 GB/s. The peak speedup was almost negligible at these bandwidths, as seen in Figure 17. The flatness of the curve indicates that bandwidth at these rates is having almost no impact on the performance and another factor (again latency, but now the memory access limit rate of 64 GB/s also a big factor) is the bottleneck.

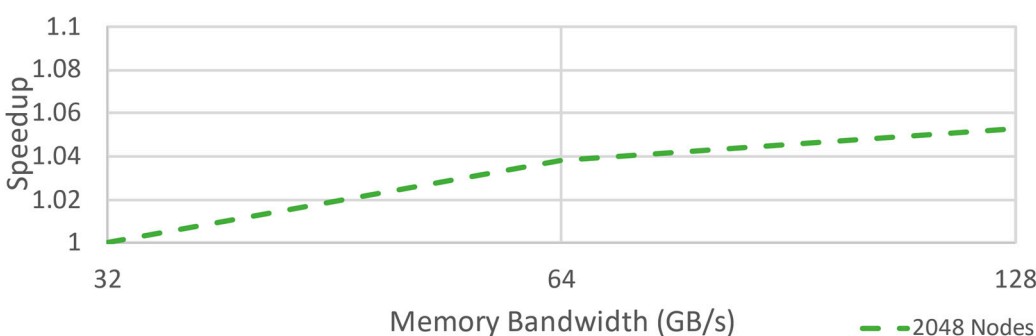

**Figure 17.** HyperX Performance for 2048 Nodes.

While the simulations in general were limited to a small number of nodes, several of the topologies (Torus, Dragonfly, and HyperX) are already known to scale well into larger applications [13]. This information, when combined with other performance analysis, can be used to jump-start design for a large-scale HPC design by focusing the efforts of future simulations to reduce the total run time required to arrive at an optimal result. This information could also support a cost-trade justification for implementation of HyperX over the common Torus topology as the performance gains are likely to more than offset the additional implementation expense.

*Related Work*

While a Fat-Tree topology is common for HPC designs, the large number of switches and high-speed links makes it less cost-effective to scale compared to other choices with similar performance characteristics [16]. A more practical solution is the Dragonfly topology, which at scale can be implemented for half the cost and closely matches the performance through an implementation that increases the effective radix of the router [14]. The HyperX topology has similar benefits and is more flexible in terms of possible configurations, able to be optimized based on physical node layout in addition to expected traffic patterns [15]. Similar research specific to optimizing the configuration of HyperX through mathematical analysis has shown how Hypercube and Flattened Butterfly topologies are a subset of the HyperX option space, but have unnecessary limitations in comparison [17].

Research quantifying the performance of individual topologies either analytically or through simulation, frequently as a function of only node count, is available in abundance [12,14,15,17,18]. Works optimizing the layout of a topology against the number of nodes [15,26] provide a solid basis for starting a network design. Comparative works between topologies evaluating performance, cost, or latency against node count in specific scenarios [11,14,16,17,19] are something this work seeks to build on. Routing algorithms for individual topologies have been analyzed with the intent of selecting the best option for the workload [9,14,15,26]. Others have also used SST to perform simulations in support of analyzing various aspects of HPC design [12,26].

**5. Conclusions**

This work has contributed by addressing a gap in existing research by evaluating the impact of variable bandwidth across topologies with the intent of identifying the point at which it no longer makes sense to invest resources into improving the bandwidth. It also presents a method for plotting performance results (speedup) not seen in other works, which simplifies the rough estimation of the optimization point; typically, raw performance numbers are shown that do not lend themselves to quick identification of return on investment. Using relative performance figures can highlight the point of diminishing returns more readily. This work has demonstrated an approach that can benefit HPC

design under cost constraints and be extended to future applications such as integrated System-on-a-Chip-based designs.

This work has also demonstrated the effectiveness of using simulations written for SST [10] to quickly and easily evaluate memory topologies for multicore HPC designs and determine the optimal configuration for specific parameters. It has demonstrated the capability to perform multi variable design trade analysis for HPC using simulation. The simulations written for this analysis are flexible and modification of any performance parameter simply involves changing a configuration variable.

The results confirm that memory bandwidth quickly stops being the limiting factor without improving the performance of other design aspects at the same time and suggest that more complex topologies are likely to provide better performance overall. The flexibility of SST [10] allows for either sequential identification and improvement of the system bottleneck or automated Monte Carlo analysis to explore the entire design space if sufficient computational resources are available to run the simulations.

The results also suggest that specific topologies may be more optimal for certain types of workloads. This is fairly intuitive, but the flexibility to change the simulated workload—even to the level of using actual executable files on predefined data sets— allows for quantitative data supporting topology selection for each HPC designs intended application. The ability of a single simulation to evaluate either general traffic patterns or very specifically defined flows with the flexibility to automate analysis of all available options is essential to the future of HPC design.

*Future Work*

One limitation of this work is its restriction to random memory access patterns for analysis; the next step would be to test additional memory access patterns to see if the results hold across workloads. Additionally, increasing the node count or taking the bandwidth beyond 8 GB/s are likely to yield additional information about the behavior of the various topologies. Both of these aspects are currently being evaluated. Making variations to the topologies or other configuration parameters is also likely to have an impact, and that design space could also be explored at some point in the future. The variables could be parameterized for Monte-Carlo analysis of an extensive trade space for a high-node-count HPC design if larger scale computing resources were available to execute the simulations. Using the capability of SST to simulate very specific software executables [10] would give a high precision performance estimation for the architecture using standard benchmarks.

**Author Contributions:** Conceptualization, J.M. and P.Z.-H.; methodology, J.M. and P.Z.-H.; software, J.M.; formal analysis, J.M.; writing—review and editing, J.M. and P.Z.-H.; supervision, P.Z.-H.; funding acquisition, P.Z.-H. All authors have read and agreed to the published version of the manuscript.

**Funding:** The APC was funded by The University of New Mexico.

**Data Availability Statement:** Numerical data for this analysis can be obtained at https://osf.io/pyfzm (accessed on 13 April 2023).

**Conflicts of Interest:** The authors declare no conflict of interest.

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
