# Peer review of "Impacts of Topology and Bandwidth on Distributed Shared Memory Systems"

_computers, doi:10.3390/computers12040086_

Round 1

Reviewer 1 Report

The paper looks interesting and is appropriate to the journal. In my opinion it requires a major revision to clarify its contribution.      The main problem with the paper is the structure, which obfuscates the contributions/arguments. To increase its research significance, there needs to be: 1) more details on how you selected the topologies 2) more references, especially because there seems to be a comparative study (if the journal allows), including for the topologies, 3) explain how the limits in the framework  (e.g. few nodes) can be generalized to HPC. 4) explain how the traffic pattern is representative of HPC. 5) explain contribution more explicitly, 6) potentially release the source code (as the framework itself seems to be the main contribution!) 7) or at the very least explain more details about the simulations (see below) 8) provide a bit more details/root cause analysis about the flattening of the speedup curve 9) better conclusions Other: .  Structural Simulation Toolkit/ HyperX is not introduced properly in the introduction. e.g. is it something existing? what is it? I think the abstract is a bit misleading, or not informative, with respect to the contribution . Introduction has super-long sentences . "build hardware prototypes of sufficient [...] variety" the meaning is simple, but the sentence is overcomplicated . I think the phrase "to this end" is used a bit improperly, at least in combination with the arguments in the paper . "seeks to directly include existing simulators" what does it mean the simulation framework seeks? directly? include? . "there is an anticipated decline in return on investment" references? anticipated by who? is this a conclusion of the paper? (if so, say so)

Author Response

Thank you for taking time to review the paper, I appreciate the level of detail in your feedback. I found your comments very clear and insightful and think addressing them has improved the paper substantially. Attached is a complete list of revisions made with a response to each comment. I welcome any further feedback you may have.

Reviewer 2 Report

The topic investigated in the work is exciting and important to discuss distributed computing design: memory network architecture.

However, the paper does provide the basic component of a scientific research paper.

The problem of the work has not clearly identified.

No related work to identify the history or previous studies in this work.

No conclusion or proposed work sections!!

Author Response

Thank you for taking the time to review this paper. Attached you will find a complete description of all the revisions made to address feedback. A copy of the responses to your specific comments is below. I welcome any further suggestions you may have towards improvement of the paper.

The problem of the work has not clearly identified.
-Description of the Problem was expanded and now has its own section to make it clear.

No related work to identify the history or previous studies in this work.
-References to related work have been more clearly identified throughout and some additional related works have been added. Most of these revisions are in the Introduction section.

No conclusion or proposed work sections!!
-The Discussion section has now been expanded and made in to 3 parts to highlight Conclusions and Future Work as their own sections.

Reviewer 3 Report

The authors of the project report "Impacts of Topology and Bandwidth on Distributed Shared Memory Systems" are mainly concerned with the design of the memory architecture. The paper is well written and provides sufficient elements of comparative optimization between different technical solutions. The various topologies are presented and briefly evaluated followed by a comparative study. The novel element is the investigation of a large number of topologies and especially the comparative study. Based on these elements I recommend the publication of the paper in this form. 

Author Response

Thank you for taking the time to review this paper. A complete description of revisions made to address feedback is attached. I took the opportunity to improve on the two areas you marked "can be improved" as listed below. I welcome any further comments you may have.

Improve the introduction to provide sufficient background and include all relevant references
-The introduction has been revised to improve this based on feedback from another reviewer.

Improve explanation for how all the cited references relevant to the research
-Added additional context for many references

Round 2

Reviewer 1 Report

The paper is significantly improved by the revision, but it still needs some additional work to make it scientifically sound. The additional work should be able to be done relatively easily.

For "1) more details on how you selected the topologies.", the new paragraph needs to provide references that show that other researchers agree with this classification/insights. A simple example is about why Fat-trees are not selected (I think you already have a reference earlier, but you need to cite it again).

"2) more references" The new conclusions section needs to be split into related work and conclusions with proper references. I see the new paragraph about related work, but it is not its place there (also inline with the other reviewer's request).

"3) explain how the limits in the framework " I disagree with the discussion that this (references for scaling and simple experiments) is enough for generalizing for scalability. If the simulator is capable for up to 512 nodes, it would be better to briefly mention numbers for 512 nodes for each case.

"4) explain how the traffic pattern is representative of HPC." I see the new comments in the text (e.g. " equivalent to the use of a properly optimized and balanced parallel HPC workload"), but they do not have a reference. I do not believe that it is representative after reading this version. Otherwise put it in the limitations/discussion.

"5) explain contribution more explicitly." the new paragraph in conclusion focuses more on related work (without references) rather than explaining the contribution. There needs to be one paragraph in "3. Description of the Problem" that explains the "multivariable optimization" which is unique to your work. Why other works have not focused on this? I think because memory bandwidth is not usually a variable, e.g. depending on the server technology.

"6) potentially release the source code" The paper is still a bit misleading on SST. The default examples from a third party tool should be mentioned a bit more carefully. I think it would be better when mentioning SST to always cite to avoid implying that this work is on SST.

Author Response

Thank you again for your review and feedback on the paper. Your comments have been addressed as described below and a complete description of changes is attached to the submission.

"1) more details on how you selected the topologies.", the new paragraph needs to provide references that show that other researchers agree with this classification/insights. A simple example is about why Fat-trees are not selected (I think you already have a reference earlier, but you need to cite it again).

-Each topology and observation now has a reference for the source.

"2) more references" The new conclusions section needs to be split into related work and conclusions with proper references. I see the new paragraph about related work, but it is not its place there (also inline with the other reviewer's request).

-The conclusions section has been split as requested and related work content moved to the new section. The new section was expanded and now contains references to 10 related works.

"3) explain how the limits in the framework " I disagree with the discussion that this (references for scaling and simple experiments) is enough for generalizing for scalability. If the simulator is capable for up to 512 nodes, it would be better to briefly mention numbers for 512 nodes for each case.

-The paper has been updated to include simulations of all topologies at larger scale. Due to time constraints for running additional simulations the scaling for the full set comparison was limited to 256 nodes (1024 cores). For HyperX only, a few simulations for 2048 nodes were able to be completed and added to the Discussion section. Observations on the differences between scales was added and figures have been added/updated accordingly. Additional information/revisions are throughout to update for this inclusion.

"4) explain how the traffic pattern is representative of HPC." I see the new comments in the text (e.g. " equivalent to the use of a properly optimized and balanced parallel HPC workload"), but they do not have a reference. I do not believe that it is representative after reading this version. Otherwise put it in the limitations/discussion.

-3 references addressing this topic have been added to section 2.1. The statement has been rephrased to incorporate the references accurately and remove the implication that it is representative of every HPC workload.

"5) explain contribution more explicitly." the new paragraph in conclusion focuses more on related work (without references) rather than explaining the contribution. There needs to be one paragraph in "3. Description of the Problem" that explains the "multivariable optimization" which is unique to your work. Why other works have not focused on this? I think because memory bandwidth is not usually a variable, e.g. depending on the server technology.

-The related work content has been removed from the conclusions and is now in the appropriate section with citations. 2 paragraphs have been added to section 3 to capture aspects of the contribution similar to other works, aspects unique from other works, likely reason why the work has not already been addressed, and why the work matters. The Conclusions section has been reworked to better identify the specific contributions and address the other additions made to the paper.

"6) potentially release the source code" The paper is still a bit misleading on SST. The default examples from a third party tool should be mentioned a bit more carefully. I think it would be better when mentioning SST to always cite to avoid implying that this work is on SST.

-Reworked lines in the Abstract, SST, Conclusions, and Materials and Methods sections to be explicit that SST is the tool used by and not the product of this work. Moved part of the SST section back into the Introduction, and the rest to a subsection of Description of the Problem. Put the simulation configuration content into a new subsection also in Description of the Problem and modified the first line for clarity. Separated SST and simulation topics under Materials and Methods into individual subsections to differentiate. Added SST as a formal citation where appropriate throughout.

Reviewer 2 Report

The paper has significantly improved.

It is clear to read, and it sounds reasonable to be accepted.

However, the quality of the figures has to be enhanced.

The related work section is significantly required to be more extensive and in a separate section. 

Author Response

Thank you again for your review and feedback on the paper. Your comments have been addressed as described below and a complete description of changes is attached to the submission.

The quality of the figures has to be enhanced.

-The size of all figures has been increased to make them easier to read. All topology diagrams have been redrawn with better attention to detail such as alignment and preventing overlap. Torus now uses color coding to identify dimensionality. Dragonfly and HyperX have been updated to better illustrate the groupings unique to the topology. Additional text has been added as appropriate for the new images. Hopefully this addresses the intent of this comment.

The related work section is significantly required to be more extensive and in a separate section.

-Related work has been moved to its own section. The section was expanded and now contains references to 10 related works.

Round 3

Reviewer 1 Report

I am happy with this revision as all feedback has been addressed carefully.